# Reproductive Tract Microbiota of Mares

**DOI:** 10.3390/vetsci11070324

**Published:** 2024-07-18

**Authors:** Ana Gil-Miranda, Jennifer Macnicol, Daniela Orellana-Guerrero, Juan C. Samper, Diego E. Gomez

**Affiliations:** 1Department of Clinical Studies, Ontario Veterinary College, University of Guelph, Guelph, ON N1G 2W1, Canada; mirandaa@uoguelph.ca (A.G.-M.); jmacnico@uoguelph.ca (J.M.); 2Veterinary Medical Teaching Hospital, University of California, Davis, CA 95616, Canada; daorellana@ucdavis.edu; 3Department of Large Animal Clinical Sciences, School of Veterinary Medicine and Biomedical Sciences, Texas A&M University, College Station, TX 77843-4475, USA; jsamper@cvm.tamu.edu

**Keywords:** microbiome, fertility, dysbiosis, endometritis, persistent breeding-induced endometritis, uterine, vaginal

## Abstract

**Simple Summary:**

Simple Summary: The female reproductive tract has a complex mix of microorganisms that are important for keeping the reproductive system healthy. When the balance of these microorganisms is disrupted, it may predispose to diseases. One such disease, endometritis, commonly causes fertility problems in mares and is difficult to diagnose and treat using standard tests. Although advanced DNA sequencing provides useful information about the microorganisms in the reproductive tract of mares, it is still challenging to define what a “normal” microbiota looks like. This review aims to summarize current knowledge about the microorganisms in the reproductive tract of mares, including the vagina, cervix, and uterus. It also explores factors that can affect these microorganisms, such as the stage of the estrous cycle, the types of bacteria, the season, and geographic location.

**Abstract:**

The female reproductive tract microbiota is a complex community of microorganisms that might be crucial in maintaining a healthy reproductive environment. Imbalances in the bacterial community (dysbiosis) and the reduction of beneficial organisms and pathogen proliferation are associated with disease. Endometritis is a common cause of fertility problems in mares, and it is still challenging to diagnose and treat based on routine culture results of certain microorganisms. Although high-throughput sequencing studies provide helpful information regarding the composition of the reproductive tract microbiota in mares, there are still challenges in defining a “normal” microbiota. The primary objective of this literature review is to summarize the current knowledge regarding the microbiota present in the reproductive tract of mares, including the vagina, cervix, and uterus. The second objective is to describe the relevant factors that can impact the reproductive microbiota of mares, including the estrous cycle stage, the type of species (genera) investigated, season, and geographic location. The rationality of identifying the normal microbiota in the reproductive tract of a mare will likely aid in understanding the impact of the microbiota on the host’s reproductive health and contribute to the treatment and prevention of equine sub and infertility issues.

## 1. Introduction

Research on the microbiota of the female reproductive tract in horses using high-throughput sequencing is a developing area of study, and little is known about it compared to other mammals. This niche has received limited attention in the equine species, particularly in comparison with other body regions, such as the gastrointestinal tract [1,2,3,4]. However, there is increasing evidence that the reproductive tracts of mammals harbour distinctive bacterial communities [5,6,7]. High-throughput sequencing studies have supported those using culture-dependent methods, suggesting that the uterus may not be a sterile environment [7]. However, the results of the culture-dependent method studies were limited, particularly in underestimating the abundance of certain bacteria, but the newer sequencing technologies allow researchers a broader identification of the bacteria present in this body system [8]. Despite this, few studies have characterized the microbiota of the uterus and vagina of healthy mares, particularly comparing both organs within the same animal. Most published studies currently use 16s rRNA sequencing to identify the bacterial microbiota, technical limitations or confounding factors can lead to discrepancies in the results, especially when determining which bacterial communities and at what proportion they are present in the uterus and vagina. It is difficult to establish a consensus on the normal microbiota of the reproductive tract in healthy mares unless sequencing is performed at a very deep level (shotgun sequencing), which is yet to be described in mares. In addition, determining a normal or core microbiota present in the reproductive tract of mares can be challenging because relevant factors such as the sampling site, the stage of the estrous cycle, age, nutritional regime, systemic disease, previous administration of antimicrobial drugs, season, and geographic location among other factors may impact the reproductive host’s health and therefore the microbiota. Therefore, this narrative literature review aims to summarize the current information regarding the microbial communities present in the reproductive tract of mares, the factors associated with their development and establishment, and their association with fertility, health, and disease.

## 2. Materials and Methods

This narrative literature review summarizes the current information regarding the microbial communities described in the reproductive tract of mares and their association with health and disease. The following keywords were used in an electronic search: mares, equine, metritis, microbiota, vaginal, uterine, and microbiota and microbiome. Articles published until January 2024 were considered for this review. Articles were collected from the following online databases: Google Scholar (https://scholar.google.ca/ (last accessed 30 January 2024), PubMed (https://pubmed.ncbi.nlm.nih.gov/) (last accessed 30 January 2024, and ScienceDirect (https://www.sciencedirect.com/) (last accessed 30 January 2024. Information from each reference was included in this review if they discussed bacterial communities present in the reproductive tract of mares, methods for detection of those bacteria, factors associated with the development and establishment of bacteria in the reproductive tract, the pathophysiology of reproductive diseases, and the relationship between microbiota and fertility.

## 3. Results and Discussion

### 3.1. Methods and Sampling Techniques to Characterize Uterine and Vaginal Microbiota

Initial studies investigating the microbial communities present in the uterus and vagina of mares were conducted using culture-based methodologies [9,10,11]. These studies reported positive cultures in 30% to 50% of the uterine samples and 40% to 95% of the vaginal swabs. In recent decades, there has been a growing interest in studying the female reproductive microbiota in horses using culture-independent methods. This approach overcomes the intrinsic limitation of culture-dependent methods in that it can detect bacteria populations that cannot be cultured and, therefore, cannot be unaccounted for otherwise [12]. For instance, the presence of dormant bacteria located deep within the endometrium (*Streptococcus zooepidemicus*) [13] and biofilm-producing bacteria (*Escherichia coli* and *Klebsiella pneumoniae*) [14] are known to promote high false negative endometritis results in routinely equine uterine cultures [15,16], due to their ability to remain in a latent state in the uterine glands of the stratum compacting [13] or live in bacterial aggregates embedded within a complex matrix capsule (extracellular polymeric substances) acquiring a layer of protection against the host immune system [17,18], challenging routine culture diagnostic [13] and treatment [19]. 

Additionally, conventional aerobic cultivation methods fail to detect known genital pathogens such as *E. coli* and *P. aeruginosa*, which have been detected using 16S ribosomal RNA sequences in mares [20]. Similarly, sequence analysis has demonstrated the presence of bacteria from both negative and positive cultures in mares during estrus and early pregnancy [21], highlighting the potential existence of bacterial microbiota in the uterus. High-throughput sequencing and bioinformatic analysis can accurately identify microbial populations in the equine reproductive tract, including live, dead, and fragmented bacterial DNA. However, experimental [15] and computational challenges can lead to high variability in sequencing results, including study design, sample handling, nucleic preparation, sequencing, quality control, assembly, binning, and functional classification [22]. Similarly, sequencing studies may have difficulty detecting bacteria beyond the family level unless sequencing is at a very deep level (shotgun sequencing) [23,24] as well as when selecting primer sets for amplification of the 16S rRNA gene, certain bacterial species may be over- or under-represented [25]. Additionally, due to the costly technique expenses, limited practitioner accessibility, and a high number of mares with reproductive problems per season, sequencing is not commonly performed in clinical settings [15]. Therefore, clinicians use relatively faster, cheaper, and less complex culture-dependent techniques for diagnosis and treatment. However, despite the limitations of both microbial detection methods, high-throughput sequencing-based methods are a robust technique that can be used to define the reproductive tract microbiota for high-quality research [12,15,26].

Most samples from the mare’s uterus for bacterial culture are obtained using guarded sterile swabs. A study by Blanchard et al. [27] compared the bacterial culture yield of samples from the endometria of 39 mares obtained using a guard with a double cannula and a distal Teflon plug, and an unguarded swab with a single, open cannula. This study showed that 38.5% and 61.5% of the samples obtained with the guarded and unguarded swabs yielded growth in blood agar at 48 h of incubation, respectively. When samples were incubated for 48 h in MacConkey’s agar, the guarded swab yielded growth in <1% of the samples, whereas 20.5% using an unguarded swab. These studies highlight the impact of different sample methods and the media on bacterial cultures from uterine samples of healthy mares. 

Similarly, the impact of sampling techniques to characterize the uterine microbiota of healthy mares has been investigated. Double-guarded sterile swabs, low-volume lavage (LVL), and endometrial biopsy produced similar results when used to characterize the uterine microbiota of 16 mares during estrous [15]. Regardless of the endometrial sampling technique, similarities in composition and relative abundance occur at the level of phyla (Proteobacteria, Firmicutes, and Bacteroidota) and genus (*Klebsiella*, *Mycoplasma*, and *Aeromonas*) level. However, these results suggested that LVL is more effective for identifying low-abundance or rare taxa than endometrial biopsy [15].

### 3.2. Uterine Microbiota

Historically, the uterus was considered a sterile environment [28]. However, studies in the 1970s and 1980s using culture-dependent approaches challenged this assumption, but conflicting results based on positive or negative bacterial cultures made it historically difficult to deduce the sterility of the uterine environment reliably. An early study by Scott et al. [9] showed that, on aerobic cultures, 33% of uterine swab specimens collected from 100 mares at slaughterhouses yielded a positive result. The most commonly identified bacteria were β-hemolytic *Streptococcus*. A later study by Hinrichs et al. [10] documented that approximately 30 and 40% of the samples from the uterus and vagina of 48 Thoroughbred and Standardbred mares with healthy reproductive tracts confirmed by endometrial biopsy and without a history of reproductive diseases yielded a positive culture on aerobic conditions, respectively. Samples were obtained using a double-guarded, occluded swab. The most common bacteria identified were *Arcanobacterium* (formerly Corynebacterium, *n* = 6), *Staphylococcus* (*n* = 7), and *Streptococcus* (*n* = 4). Purswell et al. [11] investigated the longitudinal changes in the mare’s uterus after foaling. Thirteen mares were examined at foaling and at foal heat, and seven of these mares were sampled at the second estrus using a guarded swab to obtain samples. Cultures were performed in aerobic and anaerobic conditions. During the immediate postpartum, 54% (*n* = 7) and 23% (*n* = 3) mares showed bacterial growth in aerobic and anaerobic conditions, respectively. At foal heat, 23% (*n* = 3) of the mares had a positive anaerobic bacterial culture, and no growth was reported in anaerobic conditions. At the second postpartum estrus, all 13 mares were negative in bacterial cultures from the uterus. *Streptococcus* (*n* = 14) and *Arcanobacterium* (formerly *Corynebacterium*, *n* = 4) were the most frequently isolated bacteria. These studies consistently identified *Arcanobacterium* and *Streptococcus* in the uterus of the mares suggesting that these bacteria might inhabit at different periods the uterus of healthy mares. However, the experimental design of these studies prevents determining whether those bacteria reside in the uterus and are transient visitors or invaders. 

Newer technology and results from sequencing studies have presented convincing evidence to challenge the “sterile uterus” dogma [29]. In one study, two predominant phyla (Bacteroidetes and Proteobacteria) were identified in uterine flush samples from healthy mares, suggesting that the equine uterine environment was not sterile during or after estrus [21]. The metagenomic sequencing used in this study identifies a complex population of bacteria residing within the equine uterus, regardless of the culture outcome. This study offered an initial valuable insight into the importance of implementing sophisticated diagnostic methods and sampling techniques to increase the sensitivity of the bacteriological results. The existing literature primarily reports the reproductive microbiota of healthy mares. However, despite consistently healthy sampling groups, study variations are evident. Table 1 and Figure 1 summarize the main bacterial taxa described in the uterus and vagina of healthy mares. The most abundant phyla within the equine uterus and vagina are Firmicutes, Bacteroidetes, Proteobacteria, and Actinobacteria [30,31,32]. A study by Holyoak et al. [26] showed that Firmicutes were the most abundant phylum (52%) identified in uterine samples from mares located in Louisiana, while in mares from Oklahoma, Proteobacteria (36%) and Firmicutes (36%) were the most abundant phyla detected. Proteobacteria was also the more abundant phylum detected in uterine samples from mares in Australia (40%) and from different locations from the Southern Midwestern states of the US (80%). Additionally, the study by Heil et al. [15] reported Proteobacteria (>50%) to be the most abundant phyla in the uterus of healthy mares, followed by Firmicutes and Bacteroidota regardless of the sampling method used to characterize the microbiota. Thomson et al. (2022) reported that Proteobacteria (69%), Firmicutes (21%), Bacteroidetes (7.8%), and Actinobacteria (1%) accounted for 99.6% of the total phyla abundance.

In contrast, at the genus level, any microbe has no marked dominance. *Streptococcus*, *Campylobacter*, *Klebsiella*, *Pseudomonas*, *Mycoplasma*, and *Aeromonas* are some of the genera that are consistently present within the equine reproductive microbiota, although they vary in abundance across studies [15,26,34]. Holyoak et al. [26] found that *Pseudomonas* was the most abundant (27%) genus cumulative across all samples from mares at 4 different locations, followed by *Lonsdalea* (8%), *Lactobacillus* (7.5%), *Escherichia/Shigella* (4.5%) and *Prevotella* (3%). However, *Lonsdalea* was only detected in Australian samples but not in samples of mares from Oklahoma or Louisiana. *Lactobacillus* and *Escherichia/Shigella* were dominant genera in Oklahoma or Louisiana, but their abundance in Australian mares was low. The study by Heil et al. [15] found that *Klebsiella*, *Mycoplasma*, *Aeromonas*, and *Citrobacter* were the most abundant genera present in the cervical swabs, endometrial biopsy, and low-volume lavage of healthy mares, accounting for approximately 50% of analyzed sequences. Thomson et al. reported that *Staphylococcus* (19%), *Pseudomonas* (18%), *Escherichia/Shigella* (10%), and *Klebsiella* (10%) were the most abundant genera identified in uterine biopsy of healthy mares. The reasons for these differences remain unclear because most studies focus on assessing the reproductive microbiota of a single group of animals from a single facility, all under the same environmental, housing, and nutritional conditions. Consequently, the influence of management practices cannot be evaluated within the statistical model. Similarly, studies examining the reproductive microbiota across various geographic locations with different management practices failed to report the specific conditions under which the animals were managed, preventing an assessment of these practices’ impact within the statistical model [26].

### 3.3. Vaginal Microbiota

Vaginal microbiota has also been investigated using culture-dependent methodologies. The study by Scott et al. [9] showed that 95% (95/100) of the mare’s vaginal swabs obtained at the slaughterhouse grew a bacterium in aerobic conditions. The most common bacteria identified were β-hemolytic streptococci and coliforms. The study by Hinrichs et al. [10] reported that approximately 40% (19/48) of the samples from the vagina yielded a positive culture on aerobic conditions. The most common bacteria isolated from vagina samples were *Arcanobacterium* (*n* = 10), *Streptococcus* (*n* = 4), and *Staphylococcus* (*n* = 4), while no coliforms were identified. A more recent study by Malaluang et al. [35] cultured vaginal swabs from maiden mares and mares not bred in the previous 10 years and reported that *Escherichia coli*, *Staphylococcus capitis*, *Streptococcus equisimilis*, *Streptococcus thoraltensis*, and *S. zooepidemicus* were the most common isolated bacteria with *E. coli* being dominant (40% of the samples). Malaluang et al. [36] investigated the mare’s vaginal microbiota throughout the estrous cycle from ovulation the following 3, 7, and 14 days. Samples were collected from the cranial floor of the vagina using a double-guarded occluded swab. Bacterial growth was observed in all aerobic and anaerobic cultures from the vagina of all mares on all sampling days. The number of positive cultures was higher on Days 3 and 7 (beginning and middle of diestrus), with *E. coli* and *S. zooepidemicus* being the most frequently isolated bacteria. In maiden mares, *E. coli* was particularly dominant compared to those that had foaled. 

The vaginal microbiota of healthy mares has also been studied using high-throughput sequencing approaches. Barba et al. [30] characterized the vaginal microbiota of 8 healthy Arabian mares on estrous and diestrus. Firmicutes and Bacteroides were the most abundant phylum identified in all samples regardless of the estrous cycle phase, with an abundance accounting for 32% of the sequences each, followed by Epsilonbacteraeota (9%), Actinobacteria (8%), Kiritimatiellaeota (6.5%), Proteobacteria (3.5%) and Fusobacteria (2.7%). The most abundant genera were identified at similar abundances in estrous and diestrus, with *Porphyromonas* accounting for approximately 15% of the sequences, followed by *Campylobacter* (approximately 10% of the sequences). *Corynebacterium*, *Streptococcus*, *Fusobacterium*, and *Akkermansia* were also identified in high abundance in the vagina. *Lactobacillus* only accounted for 0.18% of the abundance in estrus and 0.37% in diestrus.

The results of studies employing both culture-dependent methods and high-throughput sequencing reveal that bacteria can be detected in the uterus and vagina of healthy mares throughout the estrous cycle. Although the same bacterial phyla are identified in the uterus of healthy mares, their abundance varies among studies. At the genus level, there are significant differences between studies investigating either uterine or vaginal microbiota. These discrepancies are attributed to true differences among healthy groups, influenced by factors such as geographic location, health status, diet, body mass index, and hormone homeostasis, which can affect bacterial colonization and establishment in the mare’s uterus and vagina. Additionally, variations in the methodologies used to collect (e.g., swabs, low-volume lavage, or biopsy) and process samples may also account for differences among studies. While most current studies utilize 16S rRNA sequencing, potential technical limitations or confounding factors inherent to metagenomic analysis may further contribute to the lack of concordance, especially at lower taxonomic levels.

### 3.4. Factors Associated with Colonization and Establishment of the Reproductive Microbiota: Site, Diet, Parity, Stage of Estrous, and Species

In mares [30,37], cattle [38], women [39], female dogs [40], minipigs [41], and giant pandas [42], the vaginal and endometrial microbiota have been characterized. These studies reveal unique site-specific microbiotas. Both the canine and human endometrial microbiota demonstrate higher bacterial diversity when compared with the vaginal microbiota. However, in women, the endometrial microbiota has a higher richness than the vaginal microbiota, whereas in dogs, it is lower [5,40]. In cattle during the postpartum period, there is a shared community in the vagina and uterus [43]. The bacterial populations present in the vagina and uterus of healthy mares were presented earlier in the text, summarized in Table 1, and depicted in Figure 1. However, those studies are limited for the sample size, and more importantly, no published studies have compared the uterus and vagina microbiota within the same animal. Characterizing the vaginal and uterine microbiota in the same animal can aid in understanding how microbiota can reach the uterus [44]. Traditionally, it has been suggested that the cervical mucus plug prevents vaginal bacteria from reaching the uterus. However, this mucus plug appears to be permeable to vaginal bacteria, which may allow bacterial translocation from the proximity to the vagina [45,46]. Additionally, in women, uterine peristaltic contractions, such as those that aid in sperm transport and intensify during the follicular phase of the menstrual cycle, can also move particles into the uterus, potentially seeding the uterus with bacteria during specific phases of the estrous cycle [46,47]. Therefore, studies examining the uterine and vaginal microbiota throughout the estrous cycle in the same animal can help to identify the factors influencing and the dynamics of bacterial communities in the reproductive tract of mares. These types of studies can also aid in determining whether bacteria detected in the uterus of the mares are residents, transient invaders, or potential pathogens causing disease. 

Hormonal fluctuations and diet are some of the factors thought to influence the reproductive microbiota in cattle and humans [48,49,50]. High progesterone level is linked to a high abundance of Proteobacteria, while low progesterone concentration is associated with low Firmicutes abundance in the vagina of cattle [49]. In contrast, estrogen tends to have little influence on the vaginal microbiota of beef heifers [51]. Additionally, low glycemic index, low-fat, and nutrient-dense diets lower human bacterial vaginosis risk [52]. Other factors include ethnicity, pregnancy, hygiene, sexual exposure, and contraceptives in women [53]. Parity affects microbiota in cows, with multiparous cows having different microbiota than primiparous cows, but primiparous cows have more bacterial diversity in the uterus than their multiparous ones [54]. Little is known about the influence of these factors on the taxonomic composition of the mares’ reproductive tract.

There is an evident gap concerning the understanding of the microbiota present in the reproductive tract at different stages of the estrus cycle. While extensive research has been conducted on humans [39,55,56], there is little knowledge of mares. Hormonal fluctuations during the menstrual cycle are one of the factors responsible for shifts in the reproductive microbiota in women [53]. The effect of the estrous cycle stage is less clear in horses. A study on Arabian mares found a consistent vaginal microbiota in both the follicular and luteal phase [30]. The uterine microbiota of mares during anestrus appears to have a higher microbial diversity and richness than during estrus [15]. High estrogen concentration in the endometrium of mares in estrus may stimulate the local immune response, resulting in a less diverse microbiome during this phase of the reproductive cycle [5].

The bacteria present in the reproductive tract in healthy conditions are influenced by the type of bacteria species being studied. *Lactobacillus* has been detected in equine [57], porcine [41], bovine and ovine [58] vaginal samples. Furthermore, *Lactobacillus* dominates the healthy vagina in women [53]. However, in the vaginal microbiota of healthy mares, a predominance of *Lactobacilli* is not evident either through culture or metagenomics analyses [30]. Similarly, a low abundance of *Lactobacillus* is common in the vaginal samples of cows and ewes [58]. This marked difference suggests that *Lactobacillus* have a different role in the reproductive tracts of mares and other investigated species in comparison to humans, where *Lactobacillus* provides a barrier to opportunistic pathogen invasion and are considered a biomarker of vaginal health [59]. Furthermore, this discrepancy emphasizes the species-specific nature of certain aspects of the reproductive microbiota. Clearly, although there is a certain overlap, unique differences between mammals and reproductive tract sites necessitate species and site-specific studies regarding the female reproductive system.

### 3.5. The Reproductive Microbiota and Disease

It is widely accepted that the presence of commensal bacteria in the reproductive tract has many benefits, including enhancing the barrier function, modulating the immune response, promoting healthy microbiota, and preventing colonization by pathogenic organisms [60]. In women, the acidic environment is generated within the vagina by the commensal bacterial species *Lactobacillus* limits colonization of the reproductive tract by opportunistic pathogens [61]. *Lactobacilli* also compete for a niche with other opportunistic bacteria, thus avoiding the overgrowth of other bacteria, which could create a problematic environment within the vagina [62,63]. In women, vulvovaginal candidiasis (vaginal yeast infection) is a common disease caused by an overgrowth of yeast within the vagina, and it is associated with alterations in the population of *Lactobacillus* and *Megasphaera* [64]. The reproductive tract of cows, including the uterus and vagina, host distinct populations of microbiota [65,66], and vaginal reductions in diversity can predispose individual animals to uterine infections [43,67]. Also, shifts in the uterine bacterial community to one characterized by a low bacterial diversity dominated by *Bacteroides*, *Porphyromonas*, and *Fusobacterum* [68] predispose the cows to disease [60]. These studies exemplified the role of bacterial communities present in the vagina and uterus in maintaining the health of the reproductive tract in different species. However, studies investigating the association between vaginal and uterine microbiota and the development of inflammatory diseases of the endometrium are lacking. 

Endometritis is a common disease in mares that causes subfertility, reduced pregnancy rates, and economic losses [69,70]. The disease can be triggered by various factors, with two primary causes being persistent breeding-induced endometritis (PBIE) and chronic inflammation associated with *Streptococcus zooepidemicus* and *E. coli* infections (CIE) [70]. PBIE occurs as an inflammatory response of the endometrium more than 48 h after mating or insemination. In healthy mares, both culture-based and culture-independent techniques frequently detect *Streptococcus* and *E. coli* in uterine and vaginal samples [9,10,30,37], suggesting that these bacteria may be normal inhabitants of the reproductive tract. However, factors that disrupt the normal balance of the vaginal and uterine microbiota (dysbiosis) could facilitate the proliferation of certain bacteria, leading to endometrial inflammation [60]. Therefore, studies evaluating the association between vaginal and uterine dysbiosis, and the development of endometritis could provide valuable insights into the pathophysiology of the disease. This knowledge could inform potential therapeutic and preventive strategies for managing endometritis beyond the use of antimicrobial drugs.

### 3.6. The Reproductive Microbiota and Fertility

In humans and other mammalian species, dysbiosis or a non-balanced state of the vaginal microbiome is consistently associated with poor reproductive outcomes, including repeated insemination frequency, low pregnancy rates, and negative in vitro fertilization rates [52]. Alterations of the normal microbiota, synergistic effects with co-existing bacteria, and microbial composition imbalances in the uterus are thought to predispose to infections known as endometritis or metritis, which affect different layers of the uterus and are a major cause of infertility in most mammalian females [13,14,71,72]. 

In healthy women, the normal vaginal microbiota is primarily composed of Firmicutes, with *Lactobacillus* being the most prevalent genera. The uterus has a more diverse microbiota, including *Lactobacillus*, *Bacteroides*, *Gardnerella*, and *Prevotella*. Both the vaginal and uterine microbiota normally share some phyla, such as Firmicutes and Actinobacteria, observed in both culture-dependent and independent research [5,39,48,53,61,73]. In cattle, the dominant phyla within the vagina and uterus mainly include Firmicutes, Proteobacteria, Bacteroides, and Actinobacteria [74]. *Streptococcus* spp., *Staphylococcus* spp., and *Bacillus* spp. have been isolated in the uterus of healthy cattle [68]. 

In both women and cattle, having a healthy and normal uterine microbiome has been linked to positive reproductive outcomes such as high implantation rates and fertility success [38,75]. However, the impact of microbiota within the reproductive tract on disease and fertility in mares is not well understood and requires further investigation. This is an important area of study as it will likely affect the reproductive health of mares, as seen in other species.

### 3.7. Limitations in Current Equine Reproductive Microbiome Literature

Few studies account for various environmental effects, including diet and management practices or recent medical conditions, before the sample collection, which could impact study results. Oral-uterine translocation has been reported in women with preterm delivery and neonatal sepsis [76], and gut-uterine translocation via the bloodstream has also been proposed in cows [77]. However, associations between reproductive tract or digestive microbiota and reproductive outcomes have not yet been made in horses and are still controversial in humans [78]. Microbiota translocation from other parts of the body might predispose horses to detrimental reproductive outcomes, as reported in other species. Furthermore, many studies attempting to characterize the equine reproductive microbiota still need to publish their data. This limits interpretation and research validity, challenging the consolidation of knowledge on this topic. Additionally, most equine reproductive microbiota studies are descriptive in nature. Finally, the current understanding of the commensal reproductive microbiota in mares is limited, as “pathogenic” bacteria have also been reported in healthy uterine samples. Determining the reproductive microbiota in healthy mares during estrus is paramount for understanding the role of the resident microorganisms in different parts of the reproductive tract and how dysbiosis can affect fertility and the overall host’s health [26].

## 4. Conclusions

In conclusion, there remains a significant amount of ambiguity surrounding the composition, function, and impact of the microbiota inhabiting the female equine reproductive tract. More research on the female reproductive microbiota has been conducted in other mammals. However, the species-specific nature of this niche environment demands caution when trying to generalize results. Several considerations, including sampling site, reproductive phase, and likely confounding factors, including season and geographic location, should be considered in future studies. The microbiota within the reproductive tract likely impacts disease and fertility status. Therefore, this area of equine research urgently requires more attention.

## Figures and Tables

**Figure 1 vetsci-11-00324-f001:**
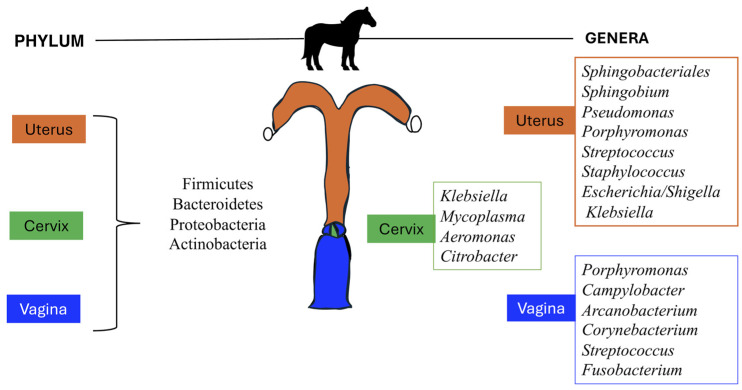
Vaginal, cervical, and uterine microbiota identified in healthy mares.

**Table 1 vetsci-11-00324-t001:** Summary of main findings obtained from studies describing the microbiota of the reproductive tract in mares.

Author(s) Objective/Hypothesis	Inclusion Criteria, (*n*), and Study’s Location, and Month	Sampling Type, Sequencing Platform, and Variable Region/Culture Medium	Main Taxa Identified	Main Results
Sathe et al., [21] * Hypothesize that the uterus of healthy mares is not sterile and is colonized by complex microflora.	Healthy mares in estrus and early pregnancy. *n* = 20. USA.	Uterine fluid. DNA sequencing of the 16S rRNA gene *** Qiime Software version 2 tm (data analysis)	Phylum: Bacteroidetes Proteobacteria Genera: Mares carrying embryos: *Sphingobacteriales* *Sphingobium* Mares not carrying embryos: *Rhodocyclaceae* and *Enterobacteriaceae.*	Pilot study illustrating that the uterus of horses is not a sterile environment during and after estrus, yet it can still achieve pregnancy in the presence of certain bacteria. Also, the study demonstrated that conventional culture methods are insufficient to identify bacteria in the uterus, which can be detected more accurately through high-throughput sequencing.
Holyoak et al., [33]	Healthy mares *n* = 29, USA.	Uterine fluid retrieved from small volume lavage (SVL). Illumina*V*4 region of the 16S rRNA gene, amplification primers: 530F and 1100R	Phylum: Proteobacteria (100%), Firmicutes (100%), Bacteroidetes (96.2%), and Actinobacteria (100%) Genera: *Pseudomonas* (100%), *Porphyromonas* (87.5%), and *Streptococcus* (61.4%).	The equine uterine microbiota is diverse, although a generalized “core” microbiota was reported in all the mares in the study, there are differences based on the animal origin.
Jones, [31] ** A. Describe and compare the vaginal, uterine, and fecal microbiota of the mare and stallion semen. B. Evaluate the impact of raw or extended semen on the uterus and vagina microbiotas following insemination.	A. Healthy mares, *n* = 16, Healthy stallion *n* = 1, USA. B. Healthy mares *n* = 8, PBIE mares (Persistent breeding-induced endometritis).	Uterine fluid from SVL and endometrial swabs collected at estrus, and 48 h post-breeding for two consecutive cycles. Illumina *V*4 region of the 16S rRNA gene, amplification primers: 515 and 926R	Phylum: Vagina/Uterus Bacteroidetes Firmicutes Actinobacteria Proteobacteria Verrucomicrobia Genera: Uterus *Corynebacterium* *Porphyromonas* *Enterobacteriaceae* *Streptococcus* Vagina Similar to the uterus, expect no *Enterobacteriaceae* but RPF12 Feces (Phylum): Bacteriodetes Firmicutes Verrucomicrobia Semen (Phylum): Actinobacteria Bacteroidetes Firmicutes	A. Feces had higher diversity than semen. Uterine and vaginal had similar diversity. All samples had unique and shared microbiotas. Sample contamination could have biased results. B. The vaginal microbiota is more dynamic than the uterine microbiota after breeding, although the dominant phyla were consistent between the two organs.
Barba et al., [30] Characterize the vaginal microbiota in Arabian mares using traditional culture-dependent and metagenomics and identify changes in estrous cycle.	Healthy mares in estrus and diestrus. *n* = 8, Spain (June–July).	Vaginal swabs. Culture-dependent: Columbia blood agar (BA), Man Rogosa Sharpe (MRS) Culture independent: Illumina *V*3/*V*4 region of the 16S rRNA gene.	Phylum: Firmicutes (100%), Bacteroidetes (100%), Proteobacteria (100%), and Actinobacteria (87.5) Genera: *Porphyromonas* (87.5%), *Campylobacter* (100%), *Arcanobacterium* (87.5%), *Corynebacterium* (87.5%), *Streptococcus* (100%), and *Fusobacterium* (87.5%).	The composition and diversity of the vaginal microbiota in Arabian mares remain consistent throughout the estrus cycle. *Lactobacillus* spp. is not dominant in the vaginal microbiota of mares.
Thomson et al., [34] Characterize the uterine microbiota in mares and predict its metabolic pathways.	Healthy mares in estrus., *n* = 21, Chile (October).	Uterine biopsy. Illumina*V*3/*V*4 region of the 16S rRNA gene, amplification primers: 341F and 785 R Positive and negative control	Phylum: Proteobacteria (69.6%), Firmicutes (21.1%), Bacteroidetes (7.8%), Actinobacteria (1.06%) Genera: *Staphylococcus* (18.88%), *Pseudomonas* (17.9%), *Escherichia*/*Shigella* (10.42%), and *Klebsiella* (9.92%).	The uterine microbiota in healthy mares is diverse, and the metabolic pathways prediction suggests that the uterus of healthy mares can produce short-chain fatty acids and amino acids.
Holyoak et al., [26] Describe the endometrial microbiome of mares in different geographical locations.	Mares with no reproductive history. *n* = 54 North America (Oklahoma, Louisiana) and Australia.	Uterine fluid retrieved by small volume lavage. IlluminaV4 region of the 16S rRNA gene, amplification primers: 515F and 806R.	Phylum: Proteobacteria (~48%), Firmicutes (30%), Bacteroidetes (12%), Actinobacteria (5%) Genera across all animals: *Pseudomonas* 27% *Lonsdalea* 8% *Lactobacillus* 7.5% *Escherichia/Shigella* 4.5% *Prevotella* 3% *Oklahoma and Louisiana Dominated by Pseudomanas* 75% *Australia (only)* *Lonsdalea* 28% Core microbiome of genera present in all samples (min abundance of 0.1%): *Lactobacillus*, *Escherichia/Shigella*, *Streptococcus*, *Blautia*, *Staphylococcus*, *Klebsiella*, *Acinetobacter*, *and Peptoanaerobacter.*	Diversity, richness, and evenness of the microbial communities of the mare’s uterus are mainly influenced by geographical location, reporting a distinct core uterine microbiome in all the mares in the study.
Heil et al., [15] Explores different sampling techniques to detect uterine microbiome in mares.	Mares in estrus without signs of endometritis on cytology and negative aerobic culture. *n* = 15, Louisiana State, USA.	Double-guarded swabs (cervix and endometrium), low-volume lavage (LVL), and endometrial biopsy Negative control; sterile unused swab (DNA isolation on same day of sample collection) IlluminaV4-V5 region of the 16S rRNA gene, amplification primers: 515F and 806R.	Phylum: Proteobacteria, Firmicutes, and Bacteroidota Genera: *Klebsiella*, *Mycoplasma*, and *Aeromonas* only. (Cervical swab: Proteobacteria, Firmicutes, Bacteroidota and Acidobacteria) *Klebsiella*, *Mycoplasma*, *Aeromonas*, and *Citrobacter.*	Alpha and beta diversity did not vary among the three sample techniques, suggesting that any method can be used for metagenomic identification in mares’ uteruses. However, LVL seems to be more efficient in sampling low-abundant or rare taxa compared to endometrial biopsy. Additionally, the cervical microbiota is more abundant than endometrial microbiota, but their compositions are similar.
Beckers et al., [32] Identify the microbiome in different sites of pregnant pony mares.	Pregnant mares (96–120 days of gestation length upon necropsy). *n* = 5, Louisiana State, USA.	Sterile swabs were collected from all sites (Placenta, vagina, anus, and oral cavity, Control-environmental swabs). Illumina *V*4 region of the 16S rRNA gene, amplification primers: 515F and 806 R	Phylum (in all sites): Firmicutes Bacteroidetes Proteobacteria Actinobacteria Genera Vagina: *Rikenellceace_RC9*, *Porphyromonas,* *Campylobacter*, and *Streptococcus*. Placenta: *Gemella*, *Rikenellaceae_RC9*, *Porphyromonas*, and *Streptococcus*.	Different richness and evenness in all samples, meaning that the microbial communities are distinct in all parts of the body tested. The placenta and oral cavity microbiome shared similarities at the genus level (*Gemella* and *Porphyromona*). Further research is needed to link the microbiome from different body sites as a biomarker of early equine placentitis.

* Abstract only; ** Thesis project (second project); *** Does not identify variable regions of the DNA sequencing.

## Data Availability

All data contained within the manuscript.

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
