# Peer review of "Reproductive Tract Microbiota of Mares"

_vetsci, 2024, doi:10.3390/vetsci11070324_

Round 1
Reviewer 1 Report
Comments and Suggestions for Authors
The manuscript discusses the composition and significance of the reproductive microbiota in healthy mares, focusing on the uterus and vagina. It highlights the variability of bacterial populations in different studies and emphasizes the need for more comprehensive research on factors influencing microbiota composition, such as geographic location, diet, and hormonal fluctuations. It also explores the role of microbiota in reproductive health, including its impact on disease susceptibility and fertility outcomes.
Concerns-
In results some sections spend considerable space recounting historical methods and findings, which, while necessary for context, could be streamlined to focus more on recent advancements and their implications.
While the text notes differences in microbiota based on geographical locations, it does not delve deeply into how environmental factors or regional practices might contribute to these differences.
The text presents newer sequencing evidence to challenge the sterile uterus dogma but does not fully integrate these findings into a cohesive narrative about how this shifts current understanding and practice.
The discussion lacks a strong link between microbiota findings and their impact on disease diagnosis and treatment.
There are several instances of redundancy, where similar points about the limitations of culture-based methods and the advantages of sequencing are repeated.
The text acknowledges study variations but does not sufficiently explore the reasons behind these variations.
Instead of listing findings from different studies separately, the text could organize them into categories based on similarities or differences in results, making it easier for readers to grasp key patterns.
Author Response
The reviewers’ comments were immensely helpful, and we appreciate such constructive feedback regarding our original submission. After addressing the issues raised, we feel the quality of the manuscript has improved greatly. Please find below our response to the reviewer’s comments.
The manuscript discusses the composition and significance of the reproductive microbiota in healthy mares, focusing on the uterus and vagina. It highlights the variability of bacterial populations in different studies and emphasizes the need for more comprehensive research on factors influencing microbiota composition, such as geographic location, diet, and hormonal fluctuations. It also explores the role of microbiota in reproductive health, including its impact on disease susceptibility and fertility outcomes.
Concerns-
In results some sections spend considerable space recounting historical methods and findings, which, while necessary for context, could be streamlined to focus more on recent advancements and their implications.
Response: Thanks for your comment. Researchers working on microbiota studies can fall into the trap of thinking that recent advances in microbiota studies are more relevant than those obtained 30 to 40 years ago using culture-based methodologies. It is important to mention that studies reporting bacterial communities in the reproductive tract of mares using culture-based methods are more clinically relevant than untargeted sequencing methods. Therefore, the authors aimed to report and compare the information regarding microbial communities in the reproductive tract of mares obtained with both methodologies. To do so, the authors took a historical approach to present the results obtained with culture-based methodologies. If the reviewer and the editor insist on changing the full approach used in this narrative review, we will consider modifying it.
While the text notes differences in microbiota based on geographical locations, it does not delve deeply into how environmental factors or regional practices might contribute to these differences.
Response: We agree with the reviewer’s comment. The authors of the current manuscript carefully reviewed the study, reporting differences in microbiota based on geographical location (Holyoak, G.R. et al., Sci Rep 2022, 12, 14790), and did not find any information in the Materials and Methods regarding the management practices, farm characteristics and regional practices. In addition, the manuscript from Holyoak et al. 2022 does not discuss the reasons for the differences in the microbial communities of the reproductive tract of mares located in different geographic locations. In addition, no other published study reports similar results or investigates management practices that can impact the reproductive tract microbial communities. Any comment or hypothesis we stated is merely speculative; therefore, we refrain from interpreting results from other studies without any information. A sentence indicating the reasons for the differences based on geographical location remains to be investigated. Line 206 - 213.
The text presents newer sequencing evidence to challenge the sterile uterus dogma but does not fully integrate these findings into a cohesive narrative about how this shifts current understanding and practice.
Response: We agree with the reviewer’s comment. Although there is newer sequencing evidence to challenge the sterile uterus dogma currently in human medicine, it is still considered that the uterus is sterile. Similarly, in horses, very few studies report the presence of bacterial sequences in the uterus, but none of those studies have addressed the clinical implications of these findings. Therefore, the authors of the current manuscript refrained from speculating about the clinical relevance of detecting bacterial DNA/RNA using sequencing methods beyond what is presented in the manuscript. A sentence has been added to highlight this point. Line 159-160.
The discussion lacks a strong link between microbiota findings and their impact on disease diagnosis and treatment.
Response: The authors apologize for reiterating the reasons for not speculating about findings from previous studies. Currently, neither in human nor veterinary medicine, there is a single disease in which microbiota findings have changed the approaches for diagnosis or treatment. As reported in our study, only 8-9 publications have studied the reproductive tract microbiota in mares. From those 2 were abstracts and 1 thesis. The majority, if not all, of those studies are descriptive. Therefore, very little to nothing is known regarding the impact of these findings on disease diagnosis and treatment. Two sentences were added to the discussion of the manuscript to highlight these points. Line 339 – 341 and 352 – 356.
There are several instances of redundancy, where similar points about the limitations of culture-based methods and the advantages of sequencing are repeated.
Response: Thanks for your comment. As suggested, several sentences highlighting the limitations of culture-based methods and the advantages of sequencing were removed.
The text acknowledges study variations but does not sufficiently explore the reasons behind these variations.
Response: Factors associated with developing and establishing the reproductive tract are presented in lines 263 to 325. Most of the information reported is extrapolated from humans as studies using the uterine and vaginal microbiota in mares are limited to a single group of mares managed similarly. Therefore, it is impossible to assess factors influencing the impact of management practices on the microbiota. Other studies, for example, include mares from different geographic areas managed differently, but unfortunately, they fail to assess the impact of those management practices or environmental conditions on the microbiota. Line 206 – 213
Instead of listing findings from different studies separately, the text could organize them into categories based on similarities or differences in results, making it easier for readers to grasp key patterns.
Response: Thanks for your comment and advice. The manuscript's authors acknowledge different styles of presenting the same information. In this case, a historical approach combined with presenting data from culture- and sequencing-based methods was elected to write the manuscript. If the reviewer and the editor insist on changing how the data is presented, the authors would consider rewriting the manuscript.
Reviewer 2 Report
Comments and Suggestions for Authors
Dear authors,
thanks for submitting the manuscript „Reproductive Tract Microbiota of Mares“. The review deals with an important topic in equine reproduction and summarizes the current knowledge of the microbiota present in reproductive tract of mares. It is interesting and well-written. However, some mistakes have to be corrected. Please find more specific comments below.
- Please consider writing bacterial species names in italics throughout the manuscript
- L. 112: missing blank
- L. 133 missing blank
- L. 135: β missing
- L. 142: remove comma
- L. 157: dot and blank are missing
- L. 160: blank missing
- L. 174: blank missing
- Table 1: Please use consistent formatting here (dots after geographical locations, italics etc.)
- L. 204: missing blank
- L. 206: β missing
- L. 207: number in brackets missing?
- L. 210: remove comma, add dot and blank
- L. 214: remove comma
- L. 228: remove 1 %-sign
- L. 231: add closing bracket
- L. 314: Why do you start writing bacterial species in italics here? They should be in italics throughout the manuscript
Comments on the Quality of English LanguagePlease proof-read the manuscript again as I am not a native speaker.
Author Response
The reviewer’s comments were immensely helpful, and we appreciate such constructive feedback regarding our original submission. After addressing the issues raised, we feel the quality of the manuscript has improved greatly. Please find below our response to the reviewer’s comments.
Thanks for submitting the manuscript „Reproductive Tract Microbiota of Mares“. The review deals with an important topic in equine reproduction and summarizes the current knowledge of the microbiota present in reproductive tract of mares. It is interesting and well-written. However, some mistakes have to be corrected. Please find more specific comments below.
- Please consider writing bacterial species names in italics throughout the manuscript
- L. 112: missing blank
Response: corrected as suggested throughout the manuscript
- L. 133 missing blank
Response: corrected as suggested throughout the manuscript
- L. 135: β missing
Response: corrected as suggested. Line 141.
- L. 142: remove comma
Response: corrected as suggested. Line 148.
- L. 157: dot and blank are missing
Response: corrected as suggested throughout the manuscript
- L. 160: blank missing
Response: corrected as suggested throughout the manuscript
- L. 174: blank missing
Response: corrected as suggested
- Table 1: Please use consistent formatting here (dots after geographical locations, italics etc.)
Response: corrected as suggested. Table 1.
- L. 204: missing blank
Response: corrected as suggested. Line 218
- L. 206: β missing
Response: corrected as suggested. Line 220.
- L. 207: number in brackets missing?
Response: corrected as suggested. Line 221.
- L. 210: remove comma, add dot and blank
Response: corrected as suggested. Line 224.
- L. 214: remove comma.
Response: corrected as suggested. Line 228
- L. 228: remove 1 %-sign
Response: corrected as suggested. Line 241.
- L. 231: add closing bracket
Response: corrected as suggested. Line 244
- L. 314: Why do you start writing bacterial species in italics here? They should be in italics throughout the manuscript
Response: corrected as suggested
Reviewer 3 Report
Comments and Suggestions for Authors
The primary objective of this literature review is to summarize the current knowledge regarding the microbiota present in the reproductive tract of mares, including the vagina, cervix, and uterus and describe the relevant factors that can impact the reproductive microbiota of mares, including the estrous cycle stage, the type of species (genera) investigated, season, and geographic location.
The material and methods section are appropriate for conducting the study as it outlines the specific databases and keywords used for the literature search, sets a clear publication date limit, and defines the criteria for including articles in the review.
I have only missed some articles which I do not know if were discarded by the authors or were not taken into account because it was not found. May you clarify this?
· Characterization of the equine placental microbial population in healthy pregnancies. Machteld van Heule, Hugo Fernando Monteiro, Ali Bazzazan, Kirsten Scoggin, Matthew Rolston, Hossam El-Sheikh Ali, Bart C. Weimer, Barry Ball, Peter Daels, Pouya Dini. Theriogenology, Volume 206, 2023, Pages 60-70, ISSN 0093-691X, https://doi.org/10.1016/j.theriogenology.2023.04.022.
· Krekeler, N., Legione, A., Perriam, W., Finan, S., Heil, B. A., Burden, C. A., McKinnon, A. O., & Marth, C. D. (2023). Association of the uterine microbiome to mare fertility. Journal of Equine Veterinary Science, 125. https://doi.org/10.1016/j.jevs.2023.104724
Some formatting errors have been highlighted in yellow, as well as some sentences that could be written more clearly.
Line 75 extra space
Line 122, low-volume lavage (LVL), follow the same acronyms throughout the article, for example
Table 1 Holyoak et al., 2018, Uterine fluid retrieved from small volume lavage (SVL). Table 1 Jones, 2019 Uterine fluid from SVL
Line 127, the sentence is unfinished
Lines 142-144, Thirteen mares were examined at foaling, at foal heat and seven of these mares were sampled at the second estrus. A guarded swab was used to obtain samples < 48 hours postpartum, at foal heat and at second postpartum estrus. (repetitive phrase, rewrite)
Line 149, At the second postpartum estrus, all mares were negative…. (add number of mares)
Line 157, missing point and space
Line 206, review
Line 277, extra space
Line 282, extra space
Line 318 (vaginal yeast infection; VYI) acronym that is not used again during the paper

Author Response
The reviewer’s comments were immensely helpful, and we appreciate such constructive feedback regarding our original submission. After addressing the issues raised, we feel the quality of the manuscript has improved greatly. Please find below our response to the reviewer’s comments.
The primary objective of this literature review is to summarize the current knowledge regarding the microbiota present in the reproductive tract of mares, including the vagina, cervix, and uterus and describe the relevant factors that can impact the reproductive microbiota of mares, including the estrous cycle stage, the type of species (genera) investigated, season, and geographic location.
The material and methods section are appropriate for conducting the study as it outlines the specific databases and keywords used for the literature search, sets a clear publication date limit, and defines the criteria for including articles in the review.
I have only missed some articles which I do not know if were discarded by the authors or were not taken into account because it was not found. May you clarify this?
- Characterization of the equine placental microbial population in healthy pregnancies. Machteld van Heule, Hugo Fernando Monteiro, Ali Bazzazan, Kirsten Scoggin, Matthew Rolston, Hossam El-Sheikh Ali, Bart C. Weimer, Barry Ball, Peter Daels, Pouya Dini. Theriogenology, Volume 206, 2023, Pages 60-70, ISSN 0093-691X, https://doi.org/10.1016/j.theriogenology.2023.04.022.
Response: Thanks for your suggestion. This scientific article was not included in the current manuscript because discussing the placental microbiota is beyond the scope of this manuscript.
- Krekeler, N., Legione, A., Perriam, W., Finan, S., Heil, B. A., Burden, C. A., McKinnon, A. O., & Marth, C. D. (2023). Association of the uterine microbiome to mare fertility. Journal of Equine Veterinary Science, 125. https://doi.org/10.1016/j.jevs.2023.104724
Response: Thanks for your suggestion. The abstract was not included because the information reported is limited and sometimes misleading.
For example, the result section states: “The richness (number of bacterial species) and evenness (distribution of bacterial numbers) of bacterial genetic sequences was compared between maiden, barren and wet mares and correlated with reproductive success.”
Based on the information provided, it is impossible to determine how the microbiota correlates with reproductive success.
The abstract states: “In some mares, conventional culture techniques failed to identify known equine genital pathogens that were detected using 16S rRNA sequencing.”
This information is misleading because it is well known that the resolution of 16S rRNA sequencing does not allow identifying bacteria with confidence beyond the family level (maybe at the genus level). Therefore, it is impossible to identify pathogenic bacteria using 16S RNA. For these reasons, we refrain from including this abstract in our review.
Some formatting errors have been highlighted in yellow, as well as some sentences that could be written more clearly.
Line 75 extra space
Response: corrected as suggested throughout the manuscript
Line 122, low-volume lavage (LVL), follow the same acronyms throughout the article, for example Table 1 Holyoak et al., 2018, Uterine fluid retrieved from small volume lavage (SVL). Table 1 Jones, 2019 Uterine fluid from SVL
Response: corrected as suggested. Table 1.
Line 127, the sentence is unfinished
Response: good catch. The sentence was completed. Line 132 – 133.
Lines 142-144, Thirteen mares were examined at foaling, at foal heat and seven of these mares were sampled at the second estrus. A guarded swab was used to obtain samples < 48 hours postpartum, at foal heat and at second postpartum estrus. (repetitive phrase, rewrite)
Response: corrected as suggested. Line 148 – 150.
Line 149, At the second postpartum estrus, all mares were negative…. (add number of mares)
Response: corrected as suggested. Line 154
Line 157, missing point and space
Response: corrected as suggested
Line 206, review
Response: corrected as suggested
Line 277, extra space
Response: corrected as suggested throughout the manuscript
Line 282, extra space
Response: corrected as suggested throughout the manuscript
Line 318 (vaginal yeast infection; VYI) acronym that is not used again during the paper
Response: corrected as suggested. Line 331
Round 2
Reviewer 1 Report
Comments and Suggestions for Authors
The authors have provided thorough explanations, making the manuscript suitable for publication. However, there are several potential improvements that could further enhance the readability of the manuscript.